# Peripheral Nervous System Involvement in Non-Primary Pediatric Cancer: From Neurotoxicity to Possible Etiologies

**DOI:** 10.3390/jcm10143016

**Published:** 2021-07-06

**Authors:** Stefano Pro, Luciana Vinti, Alessandra Boni, Angela Mastronuzzi, Martina Scilipoti, Margherita Velardi, Anna Maria Caroleo, Elisa Farina, Fausto Badolato, Iside Alessi, Giovanni Di Nardo, Andrea Carai, Massimiliano Valeriani, Antonino Reale, Pasquale Parisi, Umberto Raucci

**Affiliations:** 1Child Neurology Unit, Department of Neuroscience, Bambino Gesù Children’s Hospital, IRCCS, 00165 Rome, Italy; stefano.pro@opbg.net (S.P.); Massimiliano.valeriani@opbg.net (M.V.); 2Department of Hematology/Oncology, Gene Therapy and Hematopoietic Transplantation, Bambino Gesù Children’s Hospital, IRCCS, 00165 Rome, Italy; luciana.vinti@opbg.net (L.V.); angela.mastronuzzi@opbg.net (A.M.); annamaria.caroleo@opbg.net (A.M.C.); iside.alessi@opbg.net (I.A.); 3Department of Maternal Infantile and Urological Sciences, Sapienza University of Rome, 00161 Rome, Italy; alessandra.boni@opbg.net (A.B.); elisa.farina@opbg.net (E.F.); 4Department of Emergency, Acceptance and General Pediatrics, Bambino Gesù Children Hospital, IRCCS, 00165 Rome, Italy; martina.scilipoti@opbg.net (M.S.); antonino.reale@opbg.net (A.R.); 5Child Neurology, NESMOS Department, Faculty of Medicine and Psychology, Sant’Andrea Hospital, Sapienza University of Rome, 00189 Rome, Italy; margherita.velardi@uniroma1.it (M.V.); faustobadolato23@gmail.com (F.B.); giovannidinardo@uniroma1.it (G.D.N.); pasquale.parisi@uniroma1.it (P.P.); 6Neurosurgery Unit, Department of Neuroscience, Bambino Gesù Children’s Hospital, IRCCS, 00165 Rome, Italy; andrea.carai@opbg.net

**Keywords:** pediatric cancer, complications, peripheral nervous system, peripheral neuropathy

## Abstract

Peripheral neuropathy is a well described complication in children with cancer. Oncologists are generally well aware of the toxicity of the main agents, but fear the side effects of new drugs. As chemotherapeutic agents have been correlated with the activation of the immune system such as in Chemotherapy Induced Peripheral Neuropathy (CIPN), an abnormal response can lead to Autoimmune Peripheral Neuropathy (APN). Although less frequent but more severe, Radiation Induced Peripheral Neuropathy may be related to irreversible peripheral nervous system (PNS). Pediatric cancer patients also have a higher risk of entering a Pediatric Intensive Care Unit for complications related to therapy and disease. Injury to peripheral nerves is cumulative, and frequently, the additional stress of a malignancy and its therapy can unmask a subclinical neuropathy. Emerging risk factors for CIPN include treatment factors such as dose, duration and concurrent medication along with patient factors, namely age and inherited susceptibilities. The recent identification of individual genetic variations has advanced the understanding of physiopathological mechanisms and may direct future treatment approaches. More research is needed on pharmacological agents for the prevention or treatment of the condition as well as rehabilitation interventions, in order to allow for the simultaneous delivery of optimal cancer therapy and the mitigation of toxicity associated with pain and functional impairment. The aim of this paper is to review literature data regarding PNS complications in non-primary pediatric cancer.

## 1. Introduction

Peripheral nervous system (PNS) complications almost always appear during and immediately after chemotherapeutic treatment, while others present months or even years later. Given the growing population of childhood cancer survivors, long-term follow-up and supportive strategies will be of increasing importance to ensure a high quality of life (QOL) after childhood cancer. Neuropathy may also limit the proper administration of the chemotherapy regimen, thereby limiting its efficacy, which is why finding strategies to overcome this complication is important. Great attention should be placed on the examination of all patients exposed to neurotoxic agents so that the management of peripheral neuropathy can be initiated quickly.

Additional studies are needed to further understand the mechanisms underlying PNS involvement in pediatric cancer, to improve surveillance strategies, particularly for young children and those with central nervous system (CNS) tumors, and most importantly, to define effective treatment options that will allow the optimization of cancer treatment and the attenuation of toxicity associated with pain and functional impairment.

Chemotherapy is widely recognized as the more common cause of peripheral neuropathy in cancer patients, and neurotoxicity is the second most important cause as a dose-limiting factor of cancer treatment [1]. The recognition of neurotoxicity patterns is important both to differentiate treatment-related symptoms from cancer involvement of the nervous system and to permit assessing dose adjustment or interruption of the treatment in order to prevent further neurologic injury.

Significant expansion of the childhood cancer survivor population correlates with the enlargement of the population potentially at risk for long-term sequelae. In the last few years, genetic risk factors associated with the increase in CIPN in cancer patients were reported, in particular related to pharmacogenomics [2].

The management of PNS involvement in pediatric patients represents the most interesting challenge for the future. Presently, there are no standard protocols for preventive and therapeutic approaches, and rehabilitation strategies are limited for pain management.

The aim of this paper is to review PNS involvement ranging from pathophysiology to clinical presentation, and therapeutic options and outcomes.

### Methods

From a methodological point of view, our contribution is a review and not a systematic review. Papers published up to March 2021 were selected through a computerized literature search using PubMed and ISI Web of Science databases. We conducted a literature search and papers relevant to this review are included in the list of references.

The following terms were entered, individually or in combinations: peripheral nervous system, neuropathy, neurotoxicity, chemotherapy-induced neuropathy, autoimmune peripheral neuropathy, radiation-induced peripheral neuropathy, polyneuropathy, critical illness, enteric neurotoxicity and pediatric cancer/pediatric oncology. The literature has been selected by each author, identifying in theirown opinion the best literature to achieve the aim of the paper.

No restrictions were made on the publication date, study design, and language. A cross-reference search was carried out to identify any further relevant data.

## 2. Chemotherapy-Induced Peripheral Neuropathy (CIPN)

Chemotherapy is the core of treatment in international pediatric cancer protocols. Its toxicities, especially when acute, may interfere not just with the Quality of life (QoL) but also with the optimal delivery of treatment and daily function [3,4].

CIPN, considered just as a transient complication, is a well-recognized and quite frequent neurologic toxicity associated with specific chemotherapy, commonly used in cancer treatment [3,4].

CIPN has been described with various drugs such as vinca alkaloids, platinum compounds, taxanes, and proteasome inhibitors, whichaffect the sensory, motor and/or autonomic components of the PNS [5] (Table 1).

Traditional chemotherapy preferentially acts on cell division, resulting in DNA damage and strand breakage and interfering with DNA repair and microtubule function. For this reason, it was expected that the PNS, due to its low rate of cellular reproduction and the presence of blood-nerve barriers, would be spared injury. The clinical signs and symptoms of CIPN are caused by axonal damage in the form of a dying-back neuropathy and from damage to dorsal root ganglia cells. This selectivity of damage is probably related to the increased permeability of the blood-nerve barrier at this level [26]. Neuropathy is primarily caused by direct damage to neurons but also by indirect alteration of the surrounding microenvironment, such as localized vascular injury [27]. The role of non-neuronal cells, such as Schwann cells, is still not fully understood. Acute chemotherapy neuropathy is reported in 20–85% of children treated for acute lymphoblastic leukemia, lymphoma, CNS tumors and non-CNS solid tumors [4,28], which may present with sensory, motor, or autonomic neuron impairments [14,29]. Symptoms of CIPN may disappear on the reduction or discontinuation of the drug in question, but may also persist during therapy and continue long-term after therapy if permanent nerve damage is created [30]. Permanent damage to the structure and function of thePNS, although rarely life-threatening, affects fine motor skills, balance, mobility, endurance, and potentially Quality of Life (QoL) [31,32,33,34]. The chemotherapeutic agents responsible for CIPN in the pediatric population are those that act mainly on the microtubule [35]. Anticancer drugs are administered in combination regimens, thus the use of more than one agent makes it difficult to ascribe which is responsible and exposes the patient to additive neurologic effects [4]. It is important to know that the management of CIPN signs and symptoms, recovery, and the delayed effects of chemotherapy may vary between adult and pediatric patients. Most drugs commonly used for neuropathic pain in adult patients have not been widely studied in children. Additionally, rehabilitation therapies have not been well evaluated in children receiving CIPN-inducing drugs. Thus, both the symptoms of CIPN and methods for alleviating them might be different between these two populations [36].

### 2.1. Risk Factors

The occurrence and gravity of neurotoxicity depend on many factors:

#### 2.1.1. Treatment Factors

Dosage: Greater peripheral neurotoxicity has been reported in pediatric population receiving higher cumulative doses of vincristine (>4 g/m^2^). [3,27].

Concurrent medication: The administration of azole antifungal agents with vincristine may be present at median time lag up to 30 days [13,37]. One study has reported various patients with atypical neuropathy with hematopoietic colony stimulating factor treatment in association with vincristine [38]. Other possible pharmacokinetic interactions are reported with nifedipine, cyclosporin, carbamazepine and phenytoin. The association of triazole and imidazole antifungal agents is thought to exacerbate vincristine toxicity [13,37]. However, azole antifungals themselves are potentially neurotoxic, with interaction due to additive toxicity and not a pharmacokinetic interaction. In fact, peripheral neuropathy has been signaled in subjects exclusively receiving long-term antifungal therapy [37]. Therefore, in consideration of the literature data and regardless of the mechanism, concomitant treatment of vincristine with CYP3A4 inhibitors, in particular azole antifungals, should be avoided.

#### 2.1.2. Disease Factors

Progressive rapid weakness, resembling Guillain-Barré syndrome (GBS), has been reported in children with hematological malignancies treated with vincristine, prevailing during the induction phase [39].

#### 2.1.3. Patient Factors

Age: Discordant results have been reported on the relationship between vincristine-induced peripheral neuropathy (VIPN) and age. An increased risk of vincristine neurotoxicity has been signaled in older children [14,27,37]. Nevertheless, van de Velde et al. [40], did not confirm a clear association between age and VIPN in pediatrics. These differing results may be due to confounding comorbidities [41].

#### 2.1.4. Genetic Risk Factors

With advances in genetic sequencing, multiple genetic mutations have been discovered that may alter chemotherapy neurotoxicity profiles [2]. In particular, in the pediatric literature, research has been conducted mainly on VIPN [3]. The candidate nucleotide polymorphisms can be divided into three categories affecting vincristine metabolism (pharmacokinetics), vincristine toxicity pathways (pharmacodynamics) and SNPs in hereditary neuropathy genes [3].

Genetic alterations in pharmacokinetic pathways: The cytochrome P450 enzyme CYP3A5 “low-expressors” have a greater incidence and severity of vincristine neurotoxicity. Allelic variation in the gene for CYP3A5 results in phenotypic differences in the expression of functional enzymes [3]. In light of the importance of CYP3A5 in vincristine metabolism, several studies have focused on the nonexpressing CYP3A5 genotype (rs776746) [13,37]. Increased occurrence, severity and duration of VIPN with greater dose reductions and omissions in CYP3A5 homozygotes are reported in children. The latter patients had higher metabolite levels 1 hour after dosing and there was a significant inverse association between metabolite levels and neuropathy severity. This indicates that the reduced vincristine metabolism in patients not expressing CYP3A5 increases the VIPN. This would also explain the race difference in VIPN, as the percentage of African Americans expressing CYP3A5 is far higher than that of Caucasians (about 60% vs. 20%) [42]. However, several independent studies in pediatrics have not demonstrated associations between CYP3A5 and drug concentrations or VIPN [13,42].

Genetic polymorphism in another pharmacokinetic gene (ABCB1) has also been reported to increase neurotoxicity, which explains the difference between Caucasian and African American children [3,15,43].

Genetic alterations in pharmacodynamic pathways: A large genome wide association study established allelic variation of the CEP72 gene, involved in microtubule formation, as being significantly associated with vincristine neuropathy in children.

Interestingly, this variant is less common in African American than Caucasian individuals, providing a second plausible explanation for the inter-race difference in VIPN [13]. A further study that investigated polymorphisms in several pharmacodynamic genes also found allelic variations that may alter the risk of neuropathy [43].

However, many other pharmacokinetic and pharmacodynamic genes have been studied, and the findings require replication in other patient cohorts. The results highlighted the importance of adequate sample sizes and the precise definition of peripheral neuropathy [43].

Genetic susceptibility to hereditary neuropathy: The third group of mutations are those in hereditary neuropathy genes. Early and severe VIPN can occur as inherited underlying susceptibility in the form of a clinical and subclinical hereditary neuropathy such as Charcot-Marie-Tooth disease [3,40]. In the literature, pediatric cases presenting with unexpected severe chemotherapy induced neurotoxicity have been reported, subsequently diagnosed as having a previously unrecognized inherited neuropathy [44,45].

Assessing the impact of preexisting neuropathy on the development and severity of CIPN is controversial because patients with preexisting neuropathy are themselves excluded from clinical trials [46].

### 2.2. CIPN of Platinum Compounds

Platinum agents, above all cisplatin and carboplatin, are utilized in treatment regimens for germ-cell tumors, osteosarcoma, neuroblastoma, CNS tumors, retinoblastoma, and hepatoblastoma. Their toxicity profiles are remarkably different, provoking damage on the dorsal root ganglion and consequently a primarily sensory neuropathy, with consequent reduced sensory nerve action potentials at electroneurogram, reported years after administration [6].

Cisplatin causes reversible peripheral sensory neuropathy, characterized by numbness, tingling, paresthesias, and sometimes Lhermitte’s sign, which occurs most commonly when high cumulative doses of cisplatin are administered [8,9,10]. Symptoms may persist for months or years after the discontinuation of cisplatin. Irreversible loss of high auditory frequencies also appears to be related to a high cumulative dose of cisplatin and generally children under 5 years of age are most affected [7,47].

Cisplatin neurotoxicity may be attributed to its interference with biological enzymes in metabolic pathways. Nevertheless, the incidence and characteristics of cisplatin peripheral neuropathy in children are currently poorly described in the literature [3].

Cisplatin-induced ototoxicity is permanent and progressive, enhanced by concomitant exposure to radiation therapy [48]. The consequences of hearing loss in children are myriad and are especially impactful for patients who are treated when very young. These consequences include the impairment of speech and language acquisition, psychosocial and cognitive development, and educational and vocational achievement. Recently, sodium thiosulfate has been studied as an otoprotectant; however, although it seems to decrease the ototoxicity of cisplatin, its lack of selectivity would give it a tumor-protective property that may limit the curative effect of chemotherapy [49].

Platinum-based CIPN interferes with DNA repair mechanisms and causes DNA damage, leading to neuronal apoptosis. The impairment of the physiological replication and transcription of mtDNA, results in the synthesis of abnormal proteins, that cause abnormalities in the mitochondria [50,51].

An altered concentration of Ca^2+^ may cause the activation of gene expression of neuronal and glial cells, the alteration of membrane excitability, possible neurotransmitters release and activation of calpain which, thanks to altered proteolysis, may determine axonal degeneration [52,53]. Platinum compounds alter the activity of Na+, K+ and TRP ion channels, resulting in the hyperexcitability of peripheral neurons and induce the activation of glial cells, leading to the attraction and activation of immune cells and the release of pro-inflammatory cytokines. This results in nociceptor sensitization by the modulation of ion channel properties and hyperexcitability of peripheral neurons [54]. The failure of mitochondrial functions has been theorized to underlie what in the clinic is called “coasting” (off-therapy worsening of the symptoms) [50].

Carboplatin-based CIPN is a second generation platinum compound and it is considered less neurotoxic than cisplatin, but the frequent use combined with vincristine, makes it difficult to appreciate its contribution to neurotoxicity. Carboplatin may cause milder peripheral neuropathy than that associated with cisplatin [55] and is generally uncommon [56].

Oxaliplatin-based CIPN is not commonly used in pediatric patients except in the case of second-line therapies, for example in the GemOx regimen. In fact, Oxaliplatin is the most neurotoxic of these compounds and is the only one that also produces an acute neurotoxicity, characterized by cold-induced dysesthesias in the hands and mouth. This is likely due to its effect of transient activation on voltage-gated sodium channels of the peripheral nerves, as a result of the chelation of calcium that increases neuronal excitability [12].

This acute neurotoxicity is temporally independent of cumulative sensory toxicity, however, there is a correlation between the severity and duration of dysesthesiae and the likelihood of developing chronic sensory neuropathy. These compounds result in more prolonged neuropathic symptoms in comparison to other chemotherapy agents, presumably due to the presence of irreversible damage to the dorsal root ganglion sensory neuron [2,57].

### 2.3. CIPN of Anti-Microtubule Agents

#### 2.3.1. Vinca Alkaloids CIPN

Vinca alkaloids (vincristine, vinblastine, vindesine, and vinorelbine) act on microtubules, which causes their cytoskeletal disorganization and disorientation within axons, leading to the inhibition of the vesicle-mediated transport of neurotransmitters and axonal degeneration and denervation [13]. Vinca alkaloid exposure is related to an increased risk of motor impairment and platinum exposure is related to an augmented risk of sensory impairment [31,32]. They are often used in the pediatric population and typically cause a length- and cumulative dose-dependent neuropathy, whose incidence increases with more frequent dosing [58].

The different affinity for tubulin (decreasing in order vincristine, vinblastine, vinorelbine) might explain the distinct neurotoxicity [59]. Although vinca alkaloids have a biological effect opposite to that of taxanes, their effect on axonal transport and mitochondria function in neurons appears similar [60]. Indeed, preventing tubulin polymerization from soluble dimers into microtubules, vincristine inhibits both fast and slow axonal transport, which leads to Wallerian degeneration, altered activity of ion channels and the hyperexcitability of peripheral neurons [11].

*Vincristine* is a key component of treatment regimens for acute lymphoblastic leukemia, medulloblastoma, low-grade glioma, neuroblastoma, Wilms’ tumor, rhabdomyosarcoma, lymphoma, Ewing’s sarcoma, and retinoblastoma, and is also the agent most frequently associated with peripheral neuropathy in children with a tumor [61,62]. Manifestations comprise reduced deep tendon reflexes [14], foot and wrist drop, gait abnormalities, and muscle weakness that may be asymmetrical [16,17], neurotic pain (jaw pain, muscle cramps), paresthesias and dysesthesia. Cranial motor nerves can be affected, causing a hoarse voice, ptosis, eye movement disorders, and rarely optic neuropathy [18]. Autonomic nerve involvement may underlie constipation, paralytic ileus, and urinary retention [16,17]. In the majority of cases, these symptoms generally recover quickly if the drug is discontinued or the dose is reduced.

Neurophysiological testing shows precocious changes in nerve conduction during chemotherapy affecting approximately 25% of patients [3,13]. These alterations are mainly motor, with reductions in muscle action potentials [63,64] that may be symmetric or asymmetric, involving the lower and upper limbs [16,17]. In childhood cancer survivors, treated with multiple cycles of vincristine, a persistent sensorimotor neuropathy was evident in 20–30% of patients, suggesting that vincristine related peripheral nerve changes can be long lasting [3,31,65,66,67,68].

Vinblastine is a chemotherapy agent frequently used in pediatric regimens for low-grade gliomas, Hodgkin’s lymphoma and desmoid tumors. Despite its structural similarity to vincristine, vinblastine’s neurotoxicity is minimal and is less pronounced than that of vincristine [19].

Vinorelbine, used in childhood relapsed or refractory leukemia, Hodgkin’s lymphoma, sarcomas, and brain tumors, has a low incidence profile of peripheral neuropathy, mainly causing constipation [4].

#### 2.3.2. Taxane-Based CIPN

Taxane-based CIPN is a sensory neuropathy due to dying back axonopathy, typically length-dependent, partially reversible after treatment suspension, and reported in 11–50% of treated children [69]. Microtubules are important for the development and maintenance of neurons, and serve as a track for anterograde and retrograde axonal transport of synaptic vesicles [70,71,72]; its disruption leads to Wallerian degeneration [67] with hyperexcitability of peripheral neurons. Nevertheless, taxanes are actually scarcely utilized in childhood cancer and they are not part of the pediatric protocols used

### 2.4. CIPN of Proteasome Inhibitors

A new class of drugs, proteasome inhibitors, is being used in pediatric oncology; in particular, the crucial role is played by bortezomib, used in leukemia and certain types of lymphomas. These drugs express their actions by inhibiting proteasomes, the primary intracellular protein degradation machinery, which results in the accumulation of cytoplasmic aggregates, including neurofilaments in neuronal cells [20,21]. Bortezomib causes a dose- and length-dependent sensory axonal peripheral neuropathy. Dorsal root ganglia neuronal cell bodies are the primary target of proteasome inhibition, with peripheral nerve degeneration occurring later. The exact mechanism by which it causes neurotoxicity is not entirely clear, although it seems to play a pivotal role in the alteration of sphingolipid metabolism caused by mutations in serine palmitoyl transferase [22]. The neurotoxicity appears to be more common in adults than children and can enhance the neurotoxicity of vinorelbine or vincristine [73,74,75].

Ceramide and sphingosine-1 phosphate indeed play an important inflammatory and nociceptive action; in particular, sphingosine-1 increases neuropathic pain by the release of glutamate from the dorsal horn [76,77,78]. Bortezomib increases the production of TNF-α and IL-1β, with an increase in sphingolipid metabolism within astrocytes [79]. Other mechanisms that seem to be important include nuclear accumulations of ubiquitinated proteins, altered protein transcription in sensory ganglion neurons [80,81], the dysregulation of mitochondrial calcium homoeostasis [20] and the interference with microtubule function that leads to a decreased axonal transport [73,82]. Moreover, the blockade of nerve-growth-factor-mediated neuronal survival through the inhibition of nuclear factor jB (NFjB) might contribute to bortezomib-induced neuropathy. In addition, interfering with mitochondrial function, increases the production of ROS [73]. This leads to apoptotic changes, the hyperexcitability of peripheral neurons, the release and elevation of pro-inflammatory cytokines, and therefore to the attraction and activation of T-lymphocytes and monocytes. The new generation of proteasome inhibitors, carfilzomib and ixazomib, seems to have a lower incidence of CIPN [83].

### 2.5. Nelarabine CIPN

Nelarabine is an antimetabolite, a water-soluble pro-drug of arabinosylguanine nucleotide triphosphate, purine analogue used for the treatment of relapsed refrac-tory T-cell acute lymphoblastic leukemia and T-cell lymphoblastic lymphoma after two or more prior treatment regimens, as bridge to stem cell transplantation [23,84]. The risk of neurotoxicity may be greater in patients with a history of intrathecal che-motherapy or craniospinal irradiation [85].

Neurological complications were somnolence and neuropathy that occurred typi-cally within the first course of nelarabine therapy and is gradual at onset and reversi-ble [24].

These complications are dose-dependent [86], and mainly be related to prior che-motherapy regimen with other neurotoxic agents [24].

Large fibre peripheral neuropathy (sensory or motor) was found in PNS toxicity [25].

In literature have also been described cases of Guillaine-Barrè Syndrome after re-ceiving high-dose cytarabine in a bone marrow transplant conditioning regimen [24] and cases of irreversible paraplegia [87,88].

Older adolescents have a poor prognosis compared to younger counterparts. In fact, the 5-year overall survival for adolescents is 42–63%, while for children it is 86–89% [89].

### 2.6. CIPN Clinical Assessment

All pediatric patients exposed to neurotoxic agents during their cancer treatment should be carefully screened for early signs and symptoms of possible peripheral neuropathy.

The most widely used clinical grading scale in each group is the National Cancer Institute Common Terminology Criteria for Adverse Events, which if administered by well-trained operators, is easy to use and has good reliability [90], although it is subject to underestimation and variable inaccuracies with the age of the patient analyzed [90,91]. The modified Balis Pediatric Scale incorporates more child-specific details but has also been shown to have limited sensitivity [92].

The Total Neuropathy Score is commonly used in CIPN, with good reliability and sensitivity [90,92,93]. A validated pediatric version of the TNS and its vincristine-specific version, have been created specifically for the ages of 5–18 years [90,91,92].

Validated rating scales are not available for children younger than 5 years of age nor for children with brain and CNS tumors, with whom CIPN assessment is an additional challenge, as neurological deficits may pre-exist [4]. In general, electrodiagnostic tests reported in the medical literature do not provide additional relevant information to routine clinical management unless other diagnostic hypotheses are at stake or for research purposes [6,17,18,34,63,64]. Neurophysiologic tests typically show more extended vincristine motor axonopathy in children than in adults [67,68].

### 2.7. CIPN Clinical Neurophysiology

Neurophysiology provides evidence of nerve alterations with the possibility to detect early functional changes sometimes prior to clinical symptoms, as well as understanding the neuropathological mechanisms and organizing future prevention strategies (Table 2). Neurophysiology is particularly useful in differential diagnosis from CIPN to other neuropathies [94]. Abnormal Nerve Conduction Study (NCS) tends to be associated with large fiber size involvement (proprioception, vibratory sensation and motor activation) and well correlates with the clinical phenotype, giving the opportunity to better characterize CIPN [95]. Typical NCS abnormalities seen in CIPN are characterized from axonal damage: small or absent sensory responses, normal or slightly prolonged distal motor latency, small compound motor action potentials, normal or slightly reduced motor conduction velocity, and normal or slightly prolonged F-wave minimum latency. Typically, conduction block/temporal dispersion is not present or may disperse slightly [96].

Although some clinical studies identified patients with neuropathy purely based on their symptoms, neurophysiological exams are increasingly incorporated into CIPN assessment protocols [97]. In particular, NCS has been shown to be useful in the early stage of CIPN, identifying high-risk patients. However, in some cases, NCS does not travel parallel to the clinical course and may not change later in the course of treatment [98]. Furthermore, some clinical symptoms (particularly pain) may be seen without abnormalities in NCS [99]. Other literature data, conversely, reported a significant compound sensory nerve action potential amplitude reduction developing prior to clinical symptoms [100]. The combination of symptom and neurophysiological assessment, composite grading scales and functional measures, provides the best overall description of CIPN. Moreover, neurophysiology has shown promising application as an early surrogate biomarker for CIPN detection [100].

Primary involvement of CIPN is a sensory or sensorimotor axonal neuropathy [46] (Table 2). The current gold standard for CIPN, recommended by the International Federation of Clinical Neurophysiology, is conventional NCS [101]. To provide quantitative evidence for the prevention of CIPN and thus study its management, clinical trials that include NCS biomarkers and patient outcome are important.

### 2.8. Therapeutic Options and Prevention Approach

The study of therapeutic approaches in pediatric CIPN is extremely sparse and mostly limited to patients with vincristine-induced neuropathy [4]. Literature data reported a moderate recommendation for treatment with duloxetine.

While a number of study trials have examined potentially neuroprotective therapies for CIPN, a recent review in adults, as reported in the American Society of Clinical Oncology (ASCO) guidelines, referred to a lack of quality [102].

The benefit of duloxetine has not yet been examined with objective assessment tools such as neurophysiological studies.

Tricyclic antidepressants, pyridoxine, pyridostigmine, and a compound topical gel containing baclofen, amitriptyline, and ketamine have been proposed based on their use in other populations with neuropathic pain [102,103]. Of the many potential neuroprotective agents used in adults, the only ones that have been trialed are carbamazepine and glutamic acid for the prevention of CIPN, and intravenous immunoglobulin, pyridoxine/pyridostigmine and gabapentin for treatment, with limited evidence for benefit [3,46].

Gabapentin and pregabalin have been used in various pediatric studies of vincristine-induced neuropathy, but their efficacy has not been unequivocally established [104,105]. In the pediatric setting, dose reduction/discontinuation of treatment with the administration of another drug is often considered when VIPN presents (especially if vincristine-related), although there are no clear indications, and the choice is up to the clinician. However, vincristine is a major component of curative therapy, so it should be reinstated or the dose increased if necessary to the limits of tolerance. [40].

Non-steroidal anti-inflammatory drugs, gabapentin, amitriptyline/nortriptyline, as well as opioids are often reliable first-line agents in the management of neuropathic pain in children with cancer, but it has been concluded that randomized controlled trials are desperately needed [106]. However, also recently there are few studies investigating the role of opioids in the treatment of CIPN in children, and no standard protocols are reported in the literature for CIPN treatment [107]. Recently, however, gabapentin and tricyclic antidepressants as first-line therapy agents have been recommended. Methadone, ketamine, and lidocaine may be useful adjuvants in selected patients [108]. As of today, in pediatric settings, dose reduction or treatment interruption is often considered to prevent or treat chemotherapy-induced neuropathy [4].

Despite the fact that drug therapy can attenuate pain symptoms, research also supports the use of nonpharmacologic interventions in children with cancer who develop neuropathy [109]. Early rehabilitation and exercise in CIPN has not been fully undertaken. Physiotherapy and occupational therapy, focused on exercise to maintain strength and function, as in other neuropathic conditions, would generally be considered important [3,46].

Rehabilitation strategies for children with CIPN should focus on improving deficits (postural control, muscle weakness, fine motor skills), support the recovery of motor control, and promote regular physical activity [110]. Loss of sensory function should also be addressed with patient and family education [4].

Tomasello et al. [111] reported, in a preliminary study, that Scrambler Therapy, a non-invasive cutaneous electrostimulation device, could be a promising aid for adolescents with CIPN in pain control. Scrambler Therapy resulted in complete relief or a dramatic reduction in CIPN pain and an improvement in QoL, which is also durable in follow-up. It caused no detected side effects and can be successfully retrained.

CIPN usually progressively worsens, interfering with patients’ therapy and QoL, as well as forcing the caregiver to reduce the optimal dose or the frequency of drug administration. As a consequence, the prevention of CIPN is currently considered a crucial issue. As of today, clear guidelines for the prevention of CIPN in the child population are still missing [112]. Of the potential neuroprotective agents, the only ones that have been trialed in the pediatric population are carbamazepine, glutamic acid and amifostine for the prevention of CIPN, and intravenous immunoglobulin, pyridoxine/pyridostigmine and gabapentin for treatment, with limited evidence (Level C) for benefit [3]. However, in a recent review of VIPN (probably, the most well studied CIPN in the pediatric population) has been reported that pyridoxine and pyridostigmine may induce an improvement in symptoms while glutamic acid and glutamine may have a good preventive role [113].

A second phase III trial showed a potential otoprotective role of sodium thiosulfate against cisplatin-induced hearing loss in children with cancer [114]. Calcium and magnesium infusions for oxaliplatin-induced neuropathy have been associated with positive preliminary data, but require further investigation [115]. Poor evidence and additional data are necessary to understand the potential utility of other treatments: acupuncture, acetyl-L-carnitine, alpha-lipoic acid, L-carnosine, cryo-compression therapy, exercise, goshajinkigan, amifostine and metformin [116]. As of today, in pediatric settings, dose reduction or treatment interruption is often considered to prevent or treat CIPN [4].

### 2.9. Long-Term Outcomes

With the increasing survival of the newer therapeutic schemes, long term CIPNs are emerging.

Primarily in the case of platinum compounds and taxanes, CIPN may last several years after the completion of chemotherapy [117]. On one hand, clinicians often perceive CIPN as an acceptable and necessary side effect of life saving therapy; on the other hand, many patients judge these symptoms, which are often underappreciated by the clinicians, as having an important impact on life quality. Indeed, in severe cases, CIPN can lead to paresis, complete immobilization, significant disability and to a higher probability to fall [118,119]. Autonomic disorder does not frequently occur but can be disabling.

Up to 30% of patients in treatment with cisplatin may experience CIPN even after therapy is discontinued. Recovery can take more than a year and is usually incomplete [120]. Oxaliplatin CIPN is an important cause of treatment discontinuation; instead, its recovery is generally faster and more complete [119,121]. While platinum based peripheral neuropathy is mostly associated with sensory impairment, patients in treatment with vinca alkaloids present an increased risk of motor impairment [119,120]. Vincristine neuropathy is often dose limiting and sometimes coasting may be experienced; long-term outcomes are quite good, and symptoms are usually reversible despite recovery potentially lasting for many months [122]. In children with acute lymphoblastic leukemia, sequelae can be seen up to several years after treatment conclusion [67,68]. Vinorelbine neuropathy usually recovers after discontinuation.

Taxane treatment generally shows, with dose reduction, an improvement in symptoms. CIPN may last for months or years after completing therapy with paclitaxel, although half of patients generally get better over a period of months, often persisting with minor symptoms and no interference with daily life activities [123].

Ixabepilone-based peripheral neuropathy is the cause of treatment interruption in up to 25% of patients. Symptoms generally get better in a few months with dose reduction or with treatment interruption [124,125], and for this reason it is recommended to reduce the dose at sensory neuropathy grade 2 and stop the treatment at grade 3 [115].

Bortezomib neuropathy is reversible within 3–4 months in the majority of patients with dose reduction or drug discontinuation [21,125].

Clinicians should be aware of the dimensions that this problem can reach and of the fact that cancer survivors may need medical monitoring and treatment for a long time; knowledge of CIPN risk factors and symptoms permits the assessment of the best therapeutical scheme for each patient [126].

## 3. Autoimmune Peripheral Neuropathy (APN)

The pathogenesis of neural cytotoxicity and CIPN [127] is due to inflammation, and drug direct activity [128,129]. In particular, chemotherapy induces: an increased production and release of pro-inflammatory cytokines; an upregulation of the expression of chemokines, including CCL2 and CX3CL1—in sensory neurons, this promotes the increase in CCL1, IL-6, IL-1β, and IL-15 mRNA expression and consequently macrophage activation and infiltration of the dorsal root ganglia (DRG) in CIPN [52,130]; mitochondrial DNA damage and defects in electron transport chain proteins, leading to mitochondrial dysfunction [131] (Figure 1); and an increase in ROS within cells, which can lead to mitochondrial apoptosis, inflammation, and subsequent nerve degeneration.

ROS can also damage phospholipids, resulting in demyelination, oxidized proteins, and an increase in carbonyl by-products, which can impair antioxidant enzymes, and destroy microtubules. Intracellular ROS can also increase pro-inflammatory mediators leading to peripheral nociceptor over-excitation [132,133].

The damage of peripheral nerves exposes epitopes. As chemotherapeutic agents have been correlated with the activation of the immune system [134], an abnormal response can lead to APN (Table 3). This happens when immunologic tolerance to key antigenic sites on the myelin, axon, nodes of Ranvier or ganglionic neurons is lost. The immune response to an infection/inflammatory event can induce a cross-reaction with peripheral nerve components (myelin and axon of peripheral nerve) because of the sharing of cross-reactive epitopes (molecular mimicry) [135], leading to an acute polyneuropathy.

APN in pediatrics include [149] Guillain-Barré syndrome (GBS) and variants, such as Miller Fisher syndrome. Other APNs such as chronic inflammatory demyelinating polyneuropathy (CIDP) [150,151], multifocal motor neuropathy (MMN) [150] and paraproteinemic demyelinating polyneuropathy [151] are almost exclusively found in adults.

Guillain-Barré syndrome rarely occurs after drugs. It is the most frequent form of acquired polyneuropathy caused by demyelination; in particular, it can also be correlated with malignancies, probably due to the depression of the immune system by long-term intensive chemotherapy [152]. GBS is an immune-mediated disorder triggered by an infection/inflammatory event that on one side leads to an activation of immune system cells (such as macrophages, glial cells) and the production of proinflammatory chemokines; this induces inflammation that can result in axonal and myelin sheath damage with consequent demyelination. On the other hand, antibodies against external antigens can lead to complement fixation and may cross-react with specific gangliosides at nerve membranes and subsequently damage Schwann cells [153,154], leading again to demyelination or axonal damage or both [155]. This molecular mimicry, in combination with complement activation, leads to nerve dysfunction and symptoms of GBS. Several of these antiganglioside antibodies are frequently present (35–40% of cases) in the serum samples obtained during the acute phase and are associated with specific subtypes of GBS (anti-GM1a, anti-GM1b, anti-GD1a, and anti-GalNAc-GD1a in acute motor axonal neuropathy (AMAN), and especially anti-GQ1b in Miller Fisher syndrome) [150]. The major forms are acute inflammatory demyelinating polyradiculoneuropathy (AIDP), Miller Fisher syndrome (MFS main features are ophthalmoplegia, ataxia, and areflexia), AMAN, and acute sensorimotor axonal neuropathy (AMSAN) (Table 4). Symptoms and signs usually progress within 1 to 2 weeks. Clinically it is manifested by acute distal muscular weakness up to flaccid paralysis, with symmetrical and ascending course, a lack of reflexes and mild to moderate sensory disturbances. Few cases of GBS are associated with platinum compounds in the literature [147,148]. It is an immune-mediated, non-dose-related side effect. Platinum compounds may act as triggers of autoimmunity by inducing an elevation of proinflammatory cytokines (TNF-α and IL-6) and by enhancing anticancer immune responses, which induce an immune reaction towards myelin antigen [146].

### 3.1. Immune Checkpoint Inhibitor(ICI)-Induced APN

Immune checkpoint inhibitors (ICIs) such as ipilimumab, nivolumab, and pembrolizumab act by blocking CTLA4 and/or PD-1/PD-L1 pathways and upregulating the immune system. Normally, these receptor ligand interactions give a “turn off” message, which which orders T cells not to attack the tumor [156]. ICIs prevent CTLA-4 or PD-1 from binding to their respective receptors, and consequently inhibition signaling is blocked, thus T lymphocytes are free to kill cancer cells (Figure 2).

ICIs enhance Th1 and Th-17 cell responses and the production of cytokines (IL-6 and IL-17) that lead to abnormal T-regulatory (Treg) cell function and humoral immunity [156]. Many autoimmune diseases are related to an altered Treg/Th17 cell axis. Demyelination is the primary underlying mechanism of neuropathy following ICI therapy. Described side effects of ICIs [157] are: myasthenia gravis (anti-MuSK negative) in 2% of patients, chronic inflammatory demyelinating polyneuropathy (CIDP) (described in 36 patients to date [136,137]), sensorimotor polyneuropathy, autoimmune myopathy, Guillain-Barre syndrome (in 0.25% of patients treated with ICIs [138]) and its sometimes fatal variants [139], overlaps of MG with myositis and/or myocarditis. Other ICI-related neuromuscular complications are GBS (the second most common), Miller Fisher syndrome [140], and acute motor and sensory axonal neuropathy (AMSAN) [141].

### 3.2. Vinca Alkaloid-Induced APN

The pathogenesis of acute inflammatory demyelinating polyradiculoneuropathy in children undergoing intense chemotherapy could be related to secondary immunodepression. Immune system neoplasms can trigger acute inflammatory demyelinating polyradiculoneuropathy as some viral infections do [142]. Cases of GBS have been reported following the onset of vincristine therapy [158]; for example, a patient with acute lymphoblastic leukemia developed a fulminant motor polyradiculoneuropathy resembling an axonal variant of GBS after a few weeks of vincristine therapy [158,159].

Guillain-Barré syndrome may be a possible explanation for the severe and unexpected quadriparesis that may occur in patients with acute leukemia or lymphoma treated with vincristine [160].

Differential diagnosis between vinca alkaloid neurotoxicity and acute inflammatory demyelinating polyradiculoneuropathy can be made by examining nerve conduction velocity and performing a lumbar puncture (which points out albumin-cytological dissociation). Patients with Charcot-Marie-Tooth disease can express a severe and acute vincristine-induced neuropathy [43,143].

Fulminant neuropathy with severe motor involvement in association with vincristine therapy has been observed in patients with underlying Charcot-Marie-Tooth disease [161,162].

### 3.3. Proteasome Inhibitor Induced APN

Bortezomib can lead to a severe polyradiculoneuropathy, with an immune-mediated mechanism affecting the function and survival of immune cells such as lymphocytes and dendritic cells. Similarly to immunosuppressive or immunomodulating agents (such as TNFα antagonists), the damage induced by bortezomib can be related to a T-cell and humoral immune attack against peripheral nerve myelin, vasculitis-induced nerve ischemia, and inhibition of signaling support for axons [144]. There have been reported cases of demyelinating or mixed axonal-demyelinating neuropathy, with prominent motor involvement, albumin-cytological dissociation and lumbar root enhancement on MRI [145].

Chemotherapeutic agents can damage peripheric neuronal structures such as Schwann cells, myelin and axons in two ways: (1) inducing inflammation, and a consequent increase in proinflammatory cytokines and the exposition of self-epitopes; (2) the activation of the immune system against self-antigens leading to an APN. Nevertheless, further studies will clarify the exact pathogenesis and the proportion of patients affected by this chemotherapy-induced APN.

## 4. Radiation-Induced Peripheral Neuropathy (RIPN)

Radiation may cause damage to various tissues, such as the skin, lymph nodes, peripheral vessels, and peripheral nervous system [163]. Among these complications, neurological malfunctions represent a rare but impacting problem on the QoL of long-term cancer survivors that may lead to sensory and motor impairments in the extremities [164,165].

RIPN is usually irreversible and may appear many years after irradiation and its incidence will eventually increase due to the improved survival and longer life expectancy of patients treated when they were children [166].

At present, there is still a lack of important epidemiological studies and we must consider, regardless, RIPN as a rare complication of cancer treatment. RIPN may present with paresthesias, pain, loss of sensation, weakness and atrophy, which may differ depending on the amplitude of irradiated volume, the radio-sensibility of the irradiated tissue and the anatomic region involved [166].

Diagnosis is often difficult to make. In fact, it is difficult to differentiate neoplastic and radiation-induced plexopathy only from clinical features, although is possible to consider severe pain more suggestive of a neoplastic involvement than paresthesias, more likely referable to the radiation-induced lesion. Diagnosis may be guided by clues such as cutaneous and subcutaneous atrophy, radiotherapy tattoo marks, and combining extra-neurological signs (sternoclavicular osteoradionecrosis, radiation-induced cardiopathy, enteritis, or multiple basal cell skin carcinoma) [166]. Due to the lack of symptom specificity, diagnosis is based on neurological expertise, electrophysiological tests, MRI, PET scans and collaboration with the radiotherapist to determine the irradiation volume and site [166]. RIPN is currently a rare and mostly delayed complication of radiotherapy and the impact on the lives of long-surviving patients being treated for pediatric cancer is not yet well established. Clinicians need to be aware of the characteristics with which RIPN can manifest, to properly address differential diagnosis and to accurately manage symptoms.

### 4.1. Pathophysiology of RIPN

The exact pathophysiology underlying RIPN is not yet fully understood. Direct effects of radiations on Schwann cells and microvessels, causing demyelination and ischemia, seem to play an essential role in the alterations of the nerves’ environment, in the triggering of fibrosis and consequently in the onset of neuropathy [163,166]. Radiation-induced fibrosis is a dynamic process that involves fibroblast proliferation, extracellular matrix deposition, transforming growth factor ß1, connective tissue growth factor, and oxygen free radicals, varying from inflammation to sclerosis over several years, resulting in nerve compression in addition to direct axonal damage [166]. Histologic studies include in the pathophysiologic mechanisms, in addition to classical fibrosis, the formation of multiple nerve root cavernomas [167]. Factors influencing the risk and severity of RIPN in cancer survivors are not specific. Anyway, some radiotherapy-related factors have been identified such as a large total dose, large dose per fraction, large number of nerve fibers included in the irradiation field, heterogeneous distribution of high doses, and radiotherapy of previously treated areas [166].

### 4.2. Clinical Features

Brachial plexopathy: Radiation-induced brachial plexopathy rarely occurs as a moderately reversible syndrome, or much more frequently as a delayed and progressive syndrome in patients irradiated to chest or head and neck (lymphoma, Hodgkin’s lymphomas, thoracic and apical lung masses, etc.) [167,168]. Neurologic symptoms could appear from a few months up to more than 10 years later after radiotherapy (peak 2–4 years) [169]. There is an approximate correlation between the risk of delayed brachial plexopathy and the total radiation dose, establishing 56 Gy as the “threshold dose” [170]. The clinical onset of brachial plexopathy is often insidious, manifesting with paresthesia or dysesthesia, which may evolve into hypoesthesia and anesthesia, rather than with pain- and progressive motor weakness in a C6–T1 distribution, which is sometimes associated with fasciculations and amyotrophy [166]. Additionally the severity is variable, resulting in some cases of paralysis of the upper limb. This disorder may be accompanied by lymphoedema, which is generally due to high-dose radiotherapy or combined node exeresis and may cause an enhancement of the plexus compression [166].

Lumbosacral plexopathy: Post-radiation damage to the lumbosacral plexus most commonly occurs after the treatment of pelvic and testicular tumors, or tumors that involve para-aortic lymph nodes [171,172,173]. A mild and reversible plexopathy may occur a few months after radiotherapy, while a severe and delayed neuropathy may occur after 5 years of latency, presenting with slowly, progressive, asymmetric and bilateral leg weakness [173]. Additionally, in radiation-induced lumbosacral plexopathy, pain is usually absent [173].

Radiation-induced spinal cord injury occurs after extraneural paraspinal primary tumor irradiation, and less often in patients treated for spinal gliomas or who have undergone craniospinal irradiation. The most common form of radiation myelopathy is transient, usually occurring about 6 months after treatment, and manifesting with paresthesias and Lhermitte’s syndrosme. There is also a generally delayed form of severe radiation myelopathy (1–2 years after radiation therapy) that presents with numbness or dysesthesia of the legs, possibly progressing to weakness and sphincter dysfunction, usually without pain. In most patients, the neurological deficit progresses, leading in 50% of patients to paraplegia or quadriplegia, with difficult recovery [174].

### 4.3. Treatment of RIPN

Treatment options for patients with RIPN are limited and currently not satisfactory. The principal concern is to treat symptoms, as there is currently no curative strategy. The best approach always includes prevention in respect of radiotherapy dose limits. If a pain component is present, treatment with analgesics, benzodiazepines, tricyclic antidepressants and antiepileptics is generally effective; benzodiazepines and quinine may be used for paraesthesias and cramps, while carbamazepine may reduce nerve hyperexcitability [166]. Vitamins B1–B6 are often proposed for their neuroprotective effects, but there is no evidence of their efficacy in RIPN [166]. Physical therapy helps maintain function and prevent joint complications, which can exacerbate pain and restrict movement [166]. Due to vascular damage, heparin and warfarin have been used with the intent of retarding the progression of radiation fibrosis, with neurologic improvement described in a few patients [175]. Surgical neurolysis is an additional treatment option that rarely relieves motor or sensory impairments, and it is unclear whether it can slow the progression of deficits. Surgical methods have not proven useful to date in the management of RIPN [166,174]. Evidence for the benefit of hyperbaric oxygen on radiation-induced fibrosis is not clear, and the literature is populated by studies that have reported undefined complications and fail to demonstrate neurologic benefit [166,174,176]. The removal of triggers that can exacerbate it helps control the progression of RIPN, such as controlling high blood pressure, diabetes and alcohol abuse. Controlling acute inflammation with corticosteroids may also be useful in containing the extent and intensity of fibrosis, but there is a lack of objectivity regarding their ability to reduce nerve fiber fibrosis [166]. A combination of pentoxifyllin and tocopherol has been proved to be effective in reducing radiation-induced fibrosis [177], inducing symptom stabilization more than neurologic improvement [178]. A combination of pentoxifyllin and tocopherol with clodronate (Pentoclo) showed an improved outcome [179].

RIPN is currently a rare and mostly delayed complication of radiotherapy, the impact of which on the lives of long-surviving patients treated for pediatric cancer is not yet well established. Clinicians need to be aware of the characteristics with which radiation-induced neuropathy can manifest in order to properly address differential diagnosis and to accurately manage symptoms. It is auspicious that in the future more studies with large cohorts focused on ex-pediatric patients will be conducted, so that future efforts will be directed toward modulating the use of radiation therapy, ensuring the best efficacy and best QoL.

## 5. Enteric Nervous System and Chemotherapy-Induced Enteric Neurotoxicity

The enteric nervous system (ENS) comprises an intricate network of neurons distributed in two major ganglionated plexi (myenteric and submucosal) and other cells including interstitial cells of Cajal and enteric glial cells distributed along the gastrointestinal (GI) tract. The myenteric plexus is located between the circular and longitudinal layer of the muscularis externa and provides motor innervation to muscle layers of the GI tract, whereas the submucosal plexus innervates the epithelium and submucosal vessels controlling vascular tone and water and electrolyte balance [180].

Changes in the density and morphology of enteric neurons, so called enteric neuropathy, have been implicated in a wide range of GI disorders including achalasia, Hirschsprung’s disease, slow-transit constipation and chronic intestinal pseudo-obstruction [181,182]. Enteric neuropathies are emerging as key players in chemotherapy-induced GI dysfunction [183]. Significant enteric neuronal loss and functional and structural changes in myenteric neurons correlated with effects on GI motility and have been reported in animal models [184,185,186] and colonic samples of adult patients receiving chemotherapy [187]. In a mouse model, cisplatin administration significantly reduces the number of myenteric neurons in the gastric fundus and colon and is able to alter the proportion of a certain subpopulation of neurons in the myenteric plexus increasing the expression of neuronal nitric oxide synthase-immunoreactive (nNOS-IR) neurons and reducing the expression of calcitonin genre-related-immunoreactive neurons. These neuronal changes correlate with reduced upper GI and colonic transit [185,186]. Similarly, oxaliplatin (OXL) administration induces neuronal loss in the myenteric and submucosal plexus of the small and large bowel, causing a significant increase in the proportion of nNOS-IR colonic neurons, which is correlated with a reduction in muscle thickness and colonic contractility [187,188]. Interestingly, it has been found that co-treatment of OXL with BGP-15, a cytoprotectant, and an antioxidant, resveratrol, improved neuronal survival and related GI dysfunction, emphasizing the ENS as a promising therapeutic target for the prevention of chemotherapy-induced enteric neuropathy [189,190].

In a mous model, 5-Fluorouracil (5-FU) administration is associated with damage to the epithelial brush border and the loss of colonic crypts and goblet cells. McQuade et al. demonstrated that this acute inflammation was associated with the loss of excitatory and inhibitory neurons in the myenteric plexus, and that these changes were correlated with delayed GI transit and colonic dysmotility [191]. Interestingly, the inhibition of enteric gliosis by s100β blocker, pentamidine, prevented 5-FU-induced intestinal inflammation, oxidative stress, neuronal loss, enteric glia activation, and histological changes in mice [186]. In a mouse model treatment with irinotecan significantly reduces the number of myenteric neurons and increases the proportion of choline acetyltransferase (ChAT)-IR neurons and vesicular acetylcholine transporter (vAChT)-IR fibers in the myenteric plexus of the distal colon. These ENS changes correlated with increased GI transit time and diarrhea [192]. A recent study further demonstrated that following vincristine administration in rats, the proportion of nNOS-IR myenteric neurons in the distal colon was significantly increased [193].

Although no data are available in pediatric patients, during the first stages of life the intestine is outlined by an immature immune system, an altered intestinal permeability and a premature microbiota development, being more liable to different type of injuries [194]. Of note, chemotherapy-induced mucositis during an early, vulnerable period of neural plasticity could lead to long-lasting hypersensitivity that outlasts the acute inflammation [195].

## 6. Critical Illness Polyneuropathy in Pediatric Cancer

Critical illness polyneuropathy (CIP) is a rare entity in pediatric age that was reported for the first time by Bolton et al. in 1984. It represents a serious adverse event that may complicate the course of leukemia or other malignancies in pediatric patients [196]. CIP is a distal motor and sensory axonal polyneuropathy, often with additional myopathic involvement regarding severely ill patients in critical conditions, especially when they are admitted to the pediatric intensive care unit. Pediatric cancer patients have a higher risk of entering PICUs for complications related to therapy and disease, such as tumor lysis syndrome or immunosuppression and infections [197]. Risk factors of childhood CIP have not been understood; however, sepsis, asthma and transplantation may be responsible [198]. The etiology is attributable to the accumulation of neurotoxic factors with reduced microvascular circulation caused by endoneural hypoxia with distal axonopathy of both sensory and motor nerves as a result of its impairment of axonal transport and action potential generation [196,197,199]. In the case of systemic inflammatory response syndrome, edema of nerves is caused by interactions of inflammatory cytokines and adhesion molecules that cause microvascular dilatation with vascular permeability [196].

Electrophysiology features are a loss of both compound muscle and sensory nerve action potentials [197]. Pediatric CIP remains poorly described, and histopathological features have been reported with severe axonal neuropathy, but demyelination has not been described [200].

The timing of the presentation of CIP is not clearly correlated with specific leukemia therapy [197,200]. Hirabayashi et al. [201] reported a particular CIP in a 15-year-old-boy affected by acute lymphoblastic leukemia with a Bacillus cereus sepsis in the post-chemotherapeutic neutropenia phase. The authors postulated that neutrophils have a functional capacity as powerful mediators of tissue inflammation. The mortality of CIP in children seems to be less than in adults, but effective treatment of CIP at present is only rehabilitation therapy; there is no preventive therapy for CIP, but intensive insulin therapy and the maintenance of normoglycemia have been reported to reduce the incidence of CIP by 44% and overall mortality during intensive care by more than 40% [193]. The long-term outcome of pediatric CIP is unclear but has been reported with a 50% chance of complete recovery [202].

## 7. Conclusions

Peripheral neuropathy is a well described complication in pediatric cancer.

Presently, peripheral neuropathy is a challenging complication in chemotherapy-treated patients who may present with other possible causes of peripheral nerve damage when chemotherapy is administered and already shows paresthesia or dysesthesia before the start of treatment. Depending on the type of nerve damage, motor, sensory, or autonomic symptoms can be present. Neuropathies directly resulting from lymphomas are quite rare, as well as paraneoplastic neuropathy and cancer-associated vasculitic neuropathies. Complications of radiotherapy, including plexopathies, lower motor neuron syndrome, cranial nerve dysfunction and exceptional peripheral nerve tumors, have now been well reported. Oncologists are generally well aware of the toxicity of therapies, but the side effects of newer drugs are always to be feared and discovered, as illustrated by the complications reported with bortezomib. As chemotherapeutic agents have been correlated with the activation of immune systems (CIPN), an abnormal response can lead to APN. This happens when immunologic tolerance to myelin or axonal antigens is lost. APN includes acute/subacute neuropathy such as GBS and variants or chronic neuropathies such as CIDP or MMN. The incidence of APN as a chemotherapeutic side effect depends on the type of drug administered. Despite different mechanisms of immunity dysregulation and types of chemotherapeutic agents, APN typically presents with demyelination features with the exception of the few axonal variants of polyradiculoneuropathies.

Less frequent but more severe may be RIPN because radiation may cause irreversible PNS damage and also may appear years after irradiation, and its incidence will eventually increase due to the improved survival and longer life expectancy of patients treated when they were children. The severity of radiation is probably directly correlated to direct nerve damage, reactive fibrosis and the formation of multiple nerve root cavernomas.

The recommendation to screen children receiving anticancer therapy for peripheral neuropathy is essential in order to establish a correct treatment strategy. It is also important to point out that neuropathy can persist even after the end of anticancer therapy despite proper management. There is a need for further studies to improve knowledge of how inherited genetic variants contribute to the susceptibility and severity of peripheral neuropathy secondary to cancer therapy. In addition, better surveillance strategies are needed, particularly for young children and those with CNS tumors. Finally, more research is needed on pharmacological agents for the prevention or treatment of the condition, as well as rehabilitation interventions in order to give patients appropriate and effective anticancer therapy and ensure the alleviation of toxicity associated with pain and functional deterioration.

## Figures and Tables

**Figure 1 jcm-10-03016-f001:**
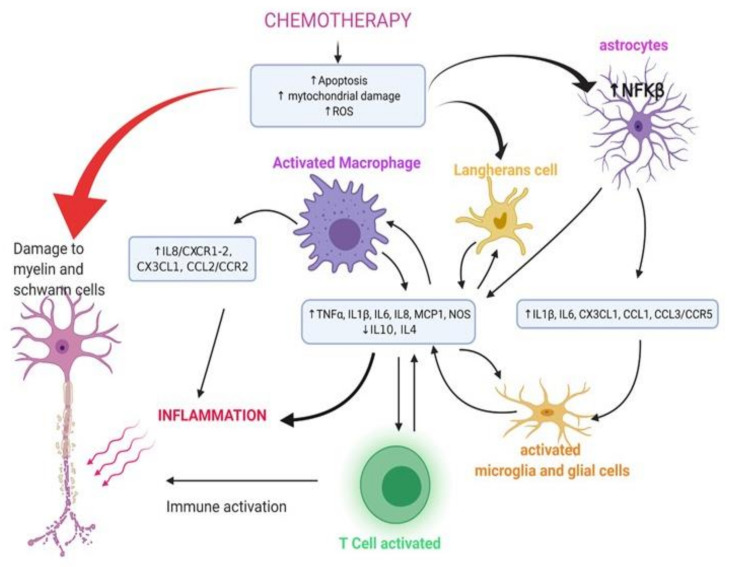
Cells and cytokines involved in chemotherapy damage (created by Biorender.com, accessed on 12 February 2021).

**Figure 2 jcm-10-03016-f002:**
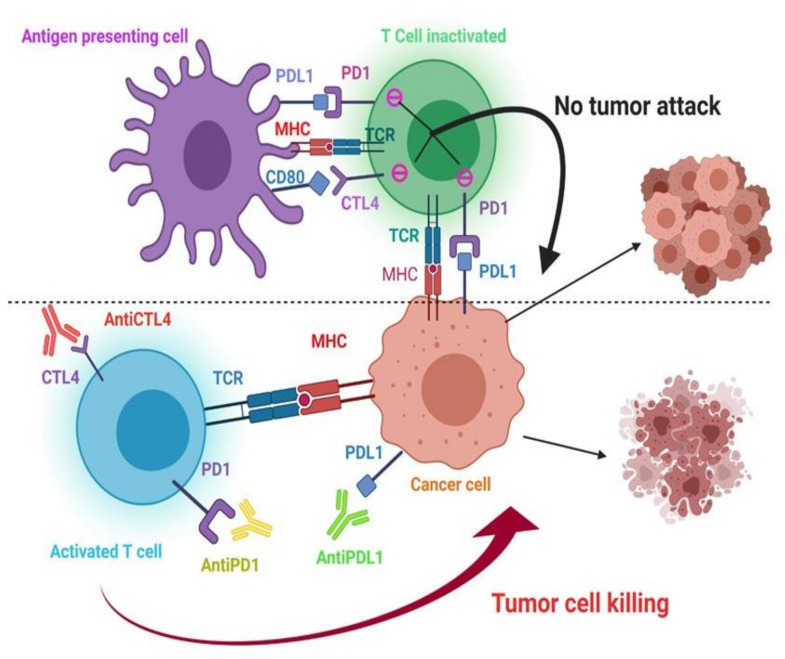
Immune checkpoint pathway PDL1: Programmed Death-Ligand 1; CTL-4: Cytotoxic T Lymphocyte Antigen 4; CD80: Cluster of Differentiation 80; MHC: Major Histocompatibility Complex; TCR: T Cell Receptor; PD1: Programmed Death 1 (Created by Biorender.com, accessed on 1 December 2020).

**Table 1 jcm-10-03016-t001:** Mechanism of action and clinical features of chemotherapeutic agents, used in pediatric protocols, in chemotherapy-induced peripheral neuropathy (CIPN).

Chemotherapeutic Agent	Mechanism of Action	Clinical Features
Platinum compounds	Damage on the dorsal root ganglion and consequently a primarily sensory neuropathy [6,7].	Cisplatin: causes reversible peripheral sensory neuropathy, characterized by numbness, tingling, and paresthesias, sometimes Lhermitte’s sign [8,9,10]. Carboplatin: milder CIPN than cisplatin [11]. Oxaliplatin: cold-induced dysesthesias in the hands and mouth [12].
Anti-microtubule agents	Vinca alkaloids: cause cytoskeletal disorganization and disorientation within axons, leading to inhibition of vesicle-mediated transport of neurotransmitters and axonal degeneration and denervation [13].	Vincristine: axonal, sensorimotor polyneuropathy, which is generally related to cumulative dose. Manifestations comprise reduced deep tendon reflexes, foot and wrist drop, gait abnormalities, and muscle weakness that may be asymmetrical neurotic pain (jaw pain, muscle cramps), paresthesias and dysesthesia. Cranial motor nerves can be affected, causing hoarse voice, ptosis, eye movement disorders, and rarely optic neuropathy. Autonomic nerve involvement may underlie constipation, paralytic ileus, and urinary retention [14,15,16,17,18]. Vinblastine and Vinorelbine: Neurotoxicity is minimal and is less pronounced than that of vincristine; sometimes constipation. If neurotoxicity is present, vincristine may be considered as an alternative chemotherapeutic drug [4,19].
Proteasome inhibitors	Degradation of intracellular proteins, resulting in accumulation of cytoplasmic aggregates, including neurofilaments in neuronal cells [20,21].	Bortezomib: causes a dose- and length-dependent sensory axonal peripheral neuropathy [22].
Nelarabine	Nelarabine is an antimetabolite, a water-soluble pro-drug of arabinosylguanine nucleotide triphosphate, a purine deoxyguanosine analog, leading to the inhibition of DNA synthesis [23]	Dose-dependent sensory and motor peripheral neuropathy; also Guillaine-Barrè Syndrome [24,25]

**Table 2 jcm-10-03016-t002:** Neurophysiological features of peripheral features from various chemotherapy agents, used in pediatric cancer protocols [46].

Chemotherapy	NCS Findings			EMG Findings
	Distal or Proximal Neuropathy	Axonal or Demyelinating Neuropathy	Sensory and/or Motor Neuropathy (S/M)	
Vincristine	Distal or Distal > Proximal;	Axonal; prolonged DML	SM or S > M	neurogenic pattern
Cisplatin	Distal; or Distal and Proximal	Axonal	S	
Oxaliplatin	In acute stage: repetitive motor discharges associated with CMAP; In chronic stage: distal S axonal			In acute stage: Fasciculations and repetitive discharges; In chronic stage: no chronic neurogenic pattern
Bortezomib		Axonal	>S or SM	
Nelarabine		Axonal; GBS-like	S or M	

**Table 3 jcm-10-03016-t003:** Autoimmune peripheral manifestations related to chemotherapeutic agents, used in pediatric cancer protocols.

Chemotherapeutic Agent	Mechanisms of Immunity Dysregulation	APN Features
Immune checkpoint inhibitors (ICIs)	ICIs enhance production of cytokines (IL-6 and IL-17) and produce an alternate Treg/Th17 with consequent abnormal T-regulatory (Treg) cell function and humoral immunity [128,129,130].	Demyelination: chronic inflammatory demyelinating polyneuropathy (CIDP), Guillain-Barrè syndrome and Miller Fisher variant. Sensorimotor polyneuropathy, autoimmune myopathy, overlaps of myasthenia gravis with myositis and/or myocarditis, acute motor and sensory axonal neuropathy (AMSAN), and myasthenia gravis (anti-MuSK negative) are also reported [136,137,138,139,140,141].
Vinka alkaloids	Immunodepression secondary to intensive chemotherapy [142].	Acute inflammatory demyelinating polyradiculoneuropathy: examining findings from nerve conduction velocity studies and performing lumbar puncture helps to differentiate between vinca alkaloid neurotoxicity and acute inflammatory demyelinating polyradiculoneuropathy [143].
Proteasome inhibitors	Both T-cell and humoral immune attack against peripheral nerve myelin, vasculitis-induced nerve ischemia, and inhibition of signaling support for axons [144].	Severe polyradiculoneuropathy. Cases of demyelinating or mixed axonal-demyelinating neuropathy, with prominent motor involvement, albumin-cytological dissociation and lumbar root enhancement on Magnetic Resonance Imaging have been reported in the literature [144,145].
Platinum Compounds	Increase in proinflammatory cytokines (TNF-α and IL-6) and enhancement of anticancer immune responses, which induce an immune reaction towards myelin antigen [146].	Anecdotal cases of Guillain-Barrè syndromes [147,148].

**Table 4 jcm-10-03016-t004:** Major form of autoimmune neuropathy and related antibodies.

Subtype	Antibodies
AMAN	Anti-GM1a/b, Anti-GD1a GalNAc-GD1
AMSAN	Anti-GM1 Anti-GD1a Anti-GM1b
AIDP	AntiGM2
MFS	Anti-GQ1b

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
