# Peer review of "Peripheral Nervous System Involvement in Non-Primary Pediatric Cancer: From Neurotoxicity to Possible Etiologies"

_jcm, 2021, doi:10.3390/jcm10143016_

Round 1

Reviewer 1 Report

I believe that the authors have adequately addressed my concerns regarding their manuscript. 

Author Response

We thank the Reviewer 1 to appreciate work done by the various Authors and for the   suggestions which have improved the paper.

Reviewer 2 Report

Authors have taken all suggestions on board. No further comments

Author Response

We thank the Reviewer 2 for Comments and Suggestions which have further improved the    paper.

This manuscript is a resubmission of an earlier submission. The following is a list of the peer review reports and author responses from that submission.

Round 1

Reviewer 1 Report

Thank you for writing this very comprehensive and excellent manuscript. It is very thorough and well done. 

My minor suggestions would be to weed through the paper and delete aspects that are not particularly relevant to pediatric literature or exceptionally rare in pedaitrics.

Otherwise no concerns. 

Author Response

We tank the Reviewer to appreciate work done by the various Authors.

We have tried to concentrate the text on pediatric aspects, even considering that in clinical practice we sometimes use adult data in the absence of pediatric ones.

Reviewer 2 Report

In this paper the author  review literature data regarding peripheral nervous system complication in non-primary pediatric cancer. From the data corrected other reports, it is important that the recommendation to screen all patients receiving cancer therapies for peripheral neuropathy and note that even after cessation of cancer therapy and with proper management is essential to prepare a correct treatment strategy.

The paper is generally clear and well written.

However, electromyography data has not shown in a patient with peripheral nervous system involvement in non-primary pediatric cancer. Electrophysiological data (latency, amplitude, nerve conduction velocity of elicited nerve compound action potential) were often used to diagnose peripheral neuropathy. 

It is important to evaluate peripheral nervous system complication.

I would recommended reconsideration after major revision: consideration of type of peripheral neuropathy with electromyography data, and description of clinical data about peripheral neuropathy.

Author Response

We thank the Reviewer to appreciate work done by the various Authors and for the suggestions which have improved the paper.

We modified the text according to the Reviewer suggestions 

Reviewer 3 Report

In this articles, the authors provided a comprehensive review of neurotoxicity affecting peripheral nervous system (PNS) from the treatment of non-primary pediatric cancer. The authors divided treatment-related neurotoxicity affecting PNS into four categories, i.e., chemotherapy induced peripheral neuropathy (CIPN), autoimmune peripheral neuropathy (APN), radiation-induced peripheral neuropathy (RIPN), chemotherapy-induced enteric neurotoxicity and critical illness polyneuropathy in pediatric cancer. The review on CIPN was most comprehensive and included risk factors, a description of CIPN related to different classes of chemotherapeutic agents, clinical assessment, neurophysiological findings, prevention, treatments and prognosis. In the review of APN, the authors first presented a description of APN-related diagnoses including Guillain Barrie Syndrome, chronic inflammatory demyelinating polyneuropathy (CIDP),  multifocal motor neuropathy (MMN), and, paraproteinaemic demyelinating polyneuropathy. They then included a description of APN related to different classes of chemotherapeutic agents. In the review of RIPN, they included pathophysiology, clinical features, and treatment of RIPN. In the review of chemotherapy-induced enteric neurotoxicity, they provided a description of the condition and putative mechanisms of the toxicity. Finally, the authors provided a brief description of critical illness polyneuropathy in pediatric cancer.

Here are my questions and comments:

  1. While the authors should be commended on their effort to provide a comprehensive review of a very important topic, a major limitation of their review is that it is unclear how much the literature reported here is related to the pediatric patient population. Since this goal of the review is to reported neurotoxicity affecting PNS among pediatric patients receiving cancer treatment, it would be important to emphasize the studies on these conditions among pediatric patients. A list of studies conducted among pediatric patients would be helpful. It is also important to point out findings from studies conducted among adult population and discuss whether these findings is applicable to pediatric patients. This should be done in every aspects of each condition discussed here including risk factors, clinical presentation, diagnostic findings, prevention, treatment and prognosis.
  2. It would be important to describe how the research was conducted. What keywords the authors used to search for literatures? How the authors achieved an exhaustive search of published studies on these topics.
  3. Many claims the authors made should be supported with relevant citations. Here are just a few examples:
    1. In line 68-69, Significant expansion of the childhood cancer survivor population correlates with the enlargement of the population potentially at risk for long-term sequelae.
    2. In line 94-95, The clinical signs symptoms due to of CIPN are caused by axonal damage in the form of a dying back neuropathy and from damage to dorsal root ganglia cells.
    3. In line 156-158, Recent advances in genetic sequencing have allowed exploration of multiple single base pair genetic mutations or, which may alter chemotherapy neurotoxicity profiles. Most of research in pediatric age, exploring genetic susceptibilities, regarded VIPN.
  4. It would be helpful to include citations in the tables.
  5. The authors provide a list of abbreviations but the list is far from complete. For example SNP in line 106 and 161, 5-FU, TLRs, ICC, EGC, etc.

Author Response

We have tried to concentrate the text on pediatric aspects, even considered that in clinical management we sometimes used adult data in the absence of pediatric ones

We thank for the suggestions. We have added a new paragraph entitled "Methods" to explain our research methodology.

We have added all suggested changes (citation in the text and in the tables, list of abbreviations) requested by the Reviewer, which have improved our paper

Round 2

Reviewer 2 Report

The paper is generally clear and well written.

There are no particular corrections.

Author Response

We thank the Reviewer to appreciate the work done by the various Authors and for the suggestions which have improved the revised form of thepaper.

Reviewer 3 Report

Among the issues I raised in my review, I pointed out that the authors should clearly identify the studies conducted among pediatric patients and those conducted among the adult population. I also asked the authors to discuss the relevance of the studies conducted among the adult population to the pediatric patients. I found that the authors's response to my comments unsatisfactory. I quickly went over the reference section of the paper and counted the studies conducted among pediatric populations. There were roughly 50 studies that met this criteria and they accounts for about 25% of the cited studies. Even though this is a limitation of the existing literature and not necessarily a limitation of this study, I believe a clear and thorough discussion on the relevance of studies conducted among adults patient to pediatric patients is warranted.

I asked the authors to provided the methods by which they identified and included the studies for this review. The authors provided a qualitative description of how the search was conducted. A PRISMA (Preferred Reporting Items for Systematic Reviews and Meta-Analyses) check list with quantitative results would have been preferred.  

Author Response

We thank the Reviewer  for Comments and Suggestions  which have further improved the re-revised text. 

We modified the text according to the required indications. In particular we have deleted various part derived from study conducted in literature among adults and no present in the pediatric population  (see text with  track changes). In fact, among the parts eliminated/changed we have substantially eliminated the parts concerning taxanes and thalidomide or similar drugs in the text and table,  not used in pediatric protocols. In addition, we modified sostantially part as autoimmune peripheral neuropathy, in  which, as rightly indicated, we have speculated almost exclusively of GBS and not of other forms found in adults. In addition, we have completely rewritten and regrouped the section 2.8 and 2.9 on preventive and therapeutic approaches based on pediatric experience. We added about 15 new references and eliminated over 50 references from the previous text, reducing the references from 244 to 195, particularly adult literaturature and older references.

We thank the reviewer for his/her comment. We specify in the new version of the manuscript that our paper is not a systematic review. We have defined a research strategy and used it in two databases (PubMed, and ISI Web of Science)  but have selected the articles at the discretion of the authors on the basis of the relationship between the drugs and the five areas of interest (i.e. chemotherapy induced peripheral neuropathy (CIPN), autoimmune peripheral neuropathy (APN), radiation-induced peripheral neuropathy (RIPN), chemotherapy-induced enteric neurotoxicity and critical illness polyneuropathy in pediatric cancer).

In particular, each author identifies in his own opinion the best literature to achieve the aim of the paper :  to review PNS involvement in non-primary pediatric cancer ranging from pathophysiology to clinical presentation, and  therapeutic options and outcomes.

From a methodological point of view, our contribution is a review and not a systematic review.